# Evaluation of Overall Survival by Restricted Mean Survival Time of Advanced Biliary Tract Cancer treated with Immunotherapy: A Systematic Review and Meta-Analysis

**DOI:** 10.3390/cancers16112077

**Published:** 2024-05-30

**Authors:** Ezequiel Mauro, Marco Sanduzzi-Zamparelli, Tamara Sauri, Alexandre Soler, Gemma Iserte, Marta Fortuny, Alejandro Forner

**Affiliations:** 1Barcelona Clinic Liver Cancer (BCLC) Group, IDIBAPS, 08036 Barcelona, Spain; msanduzzi@clinic.cat (M.S.-Z.); sauri@clinic.cat (T.S.); asolerp@clinic.cat (A.S.); giserte@clinic.cat (G.I.); mfortunyb@clinic.cat (M.F.); 2Centro de Investigación Biomédica en Red de Enfermedades Hepáticas y Digestivas (CIBERehd), 28029 Madrid, Spain; 3BCLC Group, Liver Oncology Unit, Liver Unit, ICMDM, Hospital Clinic Barcelona, Universitat de Barcelona, c/Villarroel 170. Escala 9-11, 4ª Planta, 08036 Barcelona, Spain; 4Medical Oncology Department, ICMHO, Hospital Clinic Barcelona and Translational Genomics and Targeted Therapies in Solid Tumors, IDIBAPS, 08036 Barcelona, Spain; 5Faculty of Medicine, Universitat de Barcelona, 08036 Barcelona, Spain; 6Radiology Department, CDI, Hospital Clinic Barcelona, 08036 Barcelona, Spain

**Keywords:** biliary tract cancer, immunotherapy, restricted mean survival time

## Abstract

**Simple Summary:**

This study addresses the challenge of interpreting the efficacy of immunotherapy in biliary tract cancer (BTC) in terms of overall survival (OS). Traditional methods often rely on the proportional hazard assumption, which may not hold in this context. Instead, we performed a meta-analysis of available phase 3 trials using restricted median survival time (RMST) to measure temporal benefits, providing a clearer picture of the real effects of immunotherapy-based regimens. By incorporating anti-PD-1/anti-PD-L1 agents, our analysis confirmed improvements in estimated mean OS and progression-free survival (PFS) at 24 months, as indicated by RMST, of 1.21 (95% CI: 0.49–1.93) and 1.31 (95% CI: 0.66–1.96) months, respectively. This approach not only offers a more nuanced understanding of clinical benefits, but also aids healthcare providers and patients in making informed decisions about treatment options, emphasizing a methodology that can adapt to the complexities of data.

**Abstract:**

Background: For biliary tract cancer (BTC), the addition of immunotherapy (durvalumab or pembrolizumab) to gemcitabine and cisplatin (GemCis) significantly improved overall survival (OS) in phase 3 clinical trials (RCTs). However, the interpretation and magnitude of the treatment effect is challenging because OS Kaplan–Meier curves violate the proportional hazards (PH) assumption. Analysis using restricted mean survival time (RMST) allows quantification of the benefits in the absence of PH. This systematic review and meta-analysis aims to assess the benefit of immunotherapy-based regimens for OS at 24 months using RMST analysis. Methods: A systematic review was conducted using studies published up to 8 November 2023. Only phase 3 RCTs evaluating the use of anti-PD-1/PD-L1 combined with GemCis and reporting OS were included. KM curves for OS were digitized, and the data were reconstructed. A meta-analysis for OS by RMST at 24 months was performed. Results: A total of 1754 participants from the TOPAZ-1 and KEYNOTE-966 trials were included. In TOPAZ-1, RMSTs at 24 months were 13.52 (7.92) and 12.21 (7.22) months with GemCis plus durvalumab and GemCis alone, respectively. In KEYNOTE-966, RMSTs at 24 months were 13.60 (7.76) and 12.45 (7.73) months with GemCis plus pembrolizumab and GemCis alone, respectively. Immunotherapy-based regimens showed a mean OS difference at 24 months by an RMST of 1.21 months [(95% CI: 0.49–1.93), *p* < 0.001, I^2^ = 0%]. Conclusions: Immunotherapy-based regimens improve OS in advanced BTC. Given this magnitude of benefit, it is essential to weigh up individual patient factors, preferences, and potential risks. RMST analysis provides valuable information to patients and physicians, facilitating decision-making in a value-based medical environment.

## 1. Introduction

Biliary tract cancer (BTC) constitutes less than 1% of all cancers and encompasses intrahepatic cholangiocarcinoma (iCCA), extrahepatic cholangiocarcinoma (eCCA), and gallbladder cancer (GC) [1]. Although the term BTC is used to name any tumor arising from the biliary tract, the anatomical and molecular distinctions among the different subtypes reveal that they are separate diseases in terms of therapeutic approaches and prognosis [2,3].

The prognosis for BTCs is generally poor, as the majority of patients (approximately 60%) are diagnosed with metastatic or locally advanced disease, and even in some cases of early-stage disease, surgical resection can be challenging due to the anatomical location, association with advanced chronic liver disease, and the need for significant liver parenchyma sacrifice [4]. In patients with locally advanced or metastatic BTC, gemcitabine and cisplatin (GemCis) chemotherapy has been considered the standard first-line treatment for years, showing a survival benefit with a median overall survival (OS) of approximately 12 months [5]. The benefits of GemCis have been demonstrated to be independent of age, sex, tumor stage (locally advanced or metastatic), prior treatment, and primary tumor site (iCCA, eCCA, or GC) [6].

Recent advances in the role of immunotherapy in BTC have led to a paradigm shift [4]. The publication of two positive phase 3 randomized controlled trials (RCTs), TOPAZ-1 and KEYNOTE-966, have sparked a new focus on treatment for BTC [7,8]. These trials demonstrated significant survival benefits of adding either durvalumab (TOPAZ-1) or pembrolizumab (KEYNOTE-966) to standard GemCis chemotherapy. Consequently, triple therapy comprising GemCis combined with any of these anti-PD-1/anti-PD-L1 agents has become the current standard of care in first-line therapy for BTC [9,10,11]. However, despite the positive results of the RCTs, certain factors raised concerns about the magnitude of the clinical benefit demonstrated in both studies: heterogeneity in the included populations, both in terms of geographic origin (52% of the TOPAZ-1 cohort and 45% of the KEYNOTE-966 cohort were of Asian origin) and BTC subtype, is an area of concern creating uncertainty regarding the impact of immunotherapy in this type of tumor. While a recent real-world evidence report shed light on the impact of GemCis plus durvalumab in a European population, showing similar results to the registration study, concerns have not been completely dissuaded for the widespread approval of different regulatory agencies [12,13]. Finally, while both RCTs demonstrated improved overall survival compared to GemCis, the survival curves clearly violated the assumption of proportional hazards, as seen in other immunotherapy studies, which further complicates the interpretation of the results [14,15]. 

The assumption of proportional hazards is sometimes inappropriate when Kaplan–Meier curves do not separate uniformly, making the reporting of treatment effects by hazard ratio (HR) in such situations unsuitable [16]. In this setting, alternative analytical strategies that do not rely on assumptions of proportional hazards are essential. 

Restricted mean survival time (RMST) is a well-established yet less frequently reported measure of the average time to event-free survival over a specified time interval [17]. RMST is easy to calculate; it is the area under the Kaplan–Meier curve starting from the beginning of the study and extending up to a specific time-point during follow-up. The interpreted meaning of RMST is straightforward: a larger (or smaller) value of RMST indicates an increased (or decreased) survival time compared to the control arm during the specific period [18]. The use of RMST remains a valid measure even when the assumption of proportional hazards is not met, and provides a robust alternative that not only furnishes valuable information on treatment effects, but also complements traditional measures of relative risk and absolute risk reduction [17].

In this setting, this systematic review and meta-analysis, incorporating data from phase 3 RCTs, sought to quantitatively assess the clinical advantages of combining immunotherapy regimens with GemCis. We aimed to evaluate both OS and progression-free survival (PFS) using RMST analysis. Additionally, our secondary aim was to complete a subgroup meta-analysis, considering factors such as sex, age (>65 years), geographic region, ECOG (0 or 1), disease stage (locally advanced or metastatic), and the type of BTC.

## 2. Materials and Methods

The systematic review and meta-analysis were performed in accordance with the Preferred Reporting Items for Systematic Reviews and Meta-Analyses (PRISMA) guidelines. The study has not been registered with PROSPERO.

### 2.1. Search Strategy

We systematically retrieved studies published from database inception to 8 November 2023 by searching Medline (Ovid), Embase (Elsevier), CENTRAL, and Web of Science. In addition, reference lists of reviews and meeting resources (including abstracts and posters) of the American Society of Clinical Oncology (ASCO), International Liver Cancer Association (ILCA), and European Society of Medicine Oncology (ESMO) up to 8 November 2023 were also scanned through a manual search. We searched for terms from the thesaurus and keywords being used in the title and abstract. References in eligible articles were also searched when necessary. The studies selected for review were written in English, and there were no restrictions on region, age, or follow-up duration of the participants. The full details of the search strategy are described in the Appendix A. The search process is outlined in the PRISMA schema (Appendix A) and described in Appendix B. 

### 2.2. Selection Criteria and Data Extraction

The inclusion criteria were as follows: (1) phase 3 RCTs; (2) patients diagnosed with BTC and treated with anti-PD1/PDL1 in combination with chemotherapy; and (3) studies reporting OS and PFS. The exclusion criteria were as follows: (1) anti-PD1/PDL1 combined with drugs other than chemotherapeutic agents; (2) no results provided or outcomes not relevant; and (3) duplicate studies. 

Two reviewers (EM and MSZ) independently selected potentially relevant studies based on reading the titles and abstracts. Full-text articles were then gathered and assessed for eligibility by the same 2 independent reviewers. In case of discrepancies, a consensus was reached after discussion with the senior author (AF). If a trial had more than one publication, the most recent update was selected to avoid overlapping populations.

### 2.3. Data Analysis

Two phase 3 RCTs (TOPAZ-1 [7] NCT03875235 and KEYNOTE-966 [8] NCT04003636) and an abstract presenting updated survival data from the TOPAZ-1 study at the ESMO Asia 2022 Congress [19] were selected for the analysis.

To evaluate the absolute treatment benefit of immunotherapy-based regimens combined with GemCis compared to GemCis alone in terms of OS and PFS using RMST analysis, we utilized data from the most recent exploratory analysis of TOPAZ-1, conducted on 25 February 2022, which included an additional 6.5 months of follow-up [19]. We also incorporated published data from the KEYNOTE-966 study [8].

Furthermore, OS and PFS data extracted from a Kaplan–Meier analysis in a recent multicenter Italian real-world study were included for a separate analysis to assess the consistency of OS and PFS as assessed by RMST [12].

OS and PFS curves were digitized using WebPlotDigitizer v.4.6 software [20], available at https://apps.automeris.io/wpd/ (accessed on 30 December 2023). Once digitized, the curves were processed, and the original survival data were reconstituted using the reconstructKM v.0.3.0 package [21]. To ensure data quality, survival curves were regenerated with the survival v.3.4-0 and survminer v.0.4.9 packages and the HRs were recalculated. For calculated RMSTs, a fixed time (tau) of 24 months or the last follow-up of less than 24 months was used, along with the corresponding 95% confidence intervals (95%-CI). A meta-analysis of the extracted RMST data for OS and PFS from each study was performed using the meta v.5.2-0 package. The 24-month RMSTs and their standard errors were pooled using the inverse variance method. Both common and random effects models were employed to estimate the weighted averages of the 24-month RMSTs and the mean differences between the treatment and control groups. The 95%-CI for the estimates was calculated. Heterogeneity among studies was assessed using the Cochran Q statistic and the I^2^ statistic [22]. A *p*-value < 0.05 in the Cochran Q test indicated significant heterogeneity. Furthermore, the τ2 statistic was calculated to quantify the variability between studies. The number needed to treat (NNT), with its respective 95%-CI, was calculated for each study at 24 months [23]. In addition, a sensitivity analysis was conducted for the estimated 30-month OS.

Finally, treatment effect was assessed in predefined subgroups based on factors such as sex, age (>65 years), geographic region, ECOG performance status (0 or 1), disease stage (locally advanced or metastatic), and type of BTC using standard methodology.

All analyses utilized a significance level of α = 0.05 and were conducted using R v.4.2.2.

## 3. Results

### 3.1. Study Selection and Characteristics

The search processes and the selection of the studies are described in the Appendix A.

Data from the TOPAZ-1 update and KEYNOTE-966 were included. These two studies included a total of 1754 participants. Table 1 offers comprehensive information on the inclusion and exclusion criteria of both studies, as well as the baseline characteristics and key outcomes of the included studies.

### 3.2. Assessing OS by RMST

Using data from the TOPAZ-1 study update [19], the estimated survival rates by RMST (standard deviation: SD) at 24 months were 13.52 (7.92) and 12.21 (7.22) months with GemCis plus durvalumab and GemCis alone, respectively. Similarly, in the KEYNOTE-966 study [8], these values were 13.60 (7.76) and 12.45 (7.73) months with GemCis plus pembrolizumab and GemCis alone, respectively. Figure 1 shows the variation in the RMST difference over time. In the TOPAZ-1 study, the RMST difference started near zero and gradually increased after six months, reaching approximately 60 days by the end of the 30-month period, indicating a progressively growing survival benefit of the treatment compared to the control group. Similarly, in the KEYNOTE-966 study, the RMST difference started near zero and steadily increased, reaching approximately 40 days at the end of 30 months.

After meta-analyzing both studies, the difference in the mean of the estimated survival at 24 months by RMST was 1.21 [(95%-CI: 0.49–1.93), *p* < 0.001] months longer in patients undergoing immunotherapy than in the control group. Both the fixed-effects and random-effects models showed a significant difference in the 24-month RMST for OS between the treatment and placebo groups. Furthermore, no significant heterogeneity was observed (τ^2^ = 0, I^2^ = 0%, heterogeneity test: Q = 0.05, *p* = 0.824) (Figure 2A). At this point, the NNT in TOPAZ-1 was 9 (5–71), whereas that in KEYNOTE-966 was 11 (6–59). A sensitivity analysis was conducted for the estimated 30-month OS, showing a difference in mean of 1.72 [(95%-CI: 0.83–2.60), *p* < 0.001] months in favor of patients treated with immunotherapy (Appendix A).

Finally, to assess the consistency of the OS results estimated by RMST, an additional analysis was conducted where data from registry studies and recently published data from a multicenter Italian early-access real-world study were meta-analyzed. This analysis revealed an RMST at 24 months of 13.63 [(95%CI: 13.13–14.14) heterogeneity: I^2^ = 0%, τ^2^ < 0.01, *p* = 0.523)] months in patients treated with immunotherapy (Appendix A).

### 3.3. Assessing PFS by RMST

The estimated PFS rates by RMST (SD) at 24 months in TOPAZ-1 were 8.10 (6.74) months and 6.56 (4.82) months with GemCis plus durvalumab and GemCis alone, respectively. Similarly, in the KEYNOTE-966 study at 24 months, these values were 8.51 (8.54) months and 7.49 (7.56) months with GemCis plus pembrolizumab and GemCis alone, respectively. After meta-analyzing both studies, the difference in the mean of the estimated PFS at 24 months by RMST was 1.31 [(95%-CI: 0.66–1.96), *p* < 0.001] months higher in patients undergoing immunotherapy than in the control group. No significant heterogeneity was observed (τ^2^ = 0, I^2^ = 0%, χ^2^ = 0.61, *p* = 0.435) (Figure 2B).

To assess the consistency of the PFS results estimated by RMST, an additional analysis was conducted, including data from a multicenter Italian early-access real-world study. The PFS by RMST at 24 months was 8.24 [(95%-CI: 7.82–8.65), heterogeneity: I^2^ = 0%, τ^2^ = 0, *p* = 0.669] months in patients treated with immunotherapy (Appendix A).

### 3.4. Overall Survival in the Predefined Subgroups

The meta-analysis of both studies demonstrated the OS benefit (RRs < 1) across all prespecified subgroups, including sex, age, and ECOG (0 or 1) (Figure 3).

Similarly, the results remained consistent when evaluated based on region (Asia or the rest of the world), primary tumor location (iCCA, eCCA, or GC), and stage (locally advanced or metastatic disease), although there was some heterogeneity in the analysis by tumor location and stage (Figure 4).

## 4. Discussion

Recent advances have led to the establishment of the combination of durvalumab or pembrolizumab and GemCis as the preferred first-line approach for treating locally advanced and advanced BTC, replacing GemCis as the standard therapy [9,10,11].

Our results provide a different way of interpreting the positive outcomes of the two clinical trials, focusing on the time benefit (RMST) rather than the risk scale (HR), and confirming the observed OS and PFS benefits in both RCTs. The addition of anti-PD-1/anti-PD-L1 showed a difference in the mean estimated OS and PFS values at 24 months by RMST of 1.21 (95% CI: 0.49–1.93) and 1.31 (95% CI: 0.66–1.96) months, respectively.

Another important result of the study is that the subgroup analysis showed that immunotherapy-based regimens consistently confer a survival advantage across different prespecified subgroups. However, the uncertain interactions regarding geographic diversity, especially in non-Asian populations, and BTC subtypes, in addition to the exploratory nature of the subgroup analyses, warrant careful interpretation [24]. Finally, data from the multicenter Italian early-access real-world study and the regional subgroup analysis of TOPAZ-1 indicate that the findings regarding efficacy are consistent for both OS and PFS in the non-Asian population [12,13,25].

The importance of this systematic review and meta-analysis relies on its ability to comprehensively measure potential benefits. By quantifying the effects of an intervention, the RMST difference empowers patients and clinicians to gauge whether the therapy offers a meaningful benefit, considering the patient’s values and preferences [26]. This is particularly important when treatment decisions are complicated by factors such as a limited time window for administration, potentially significant side effects, or high costs [17,27].

The use of RMST to assess survival benefits provides a valuable alternative to hazard ratios, particularly when the assumption of proportional hazards is not met [14]. It offers a practical and easily interpretable measure of survival, fostering a more comprehensive understanding of treatment effects for both healthcare providers and patients [17,18,26,27].

Currently, multiple tools are used to assess clinical benefits, both in relative and absolute terms. For instance, the ESMO-MCBS v.1.1 scale provides guidance for systematically assessing the clinical relevance of benefits [28]. This scale assigned a score of 4 (substantial magnitude of clinical benefit) to the TOPAZ-1 trial and 1 (low magnitude of clinical benefit) to the KEYNOTE-966 trial, indicating clear discordance in terms of potential clinical benefit between the two trials. The distinctions of 4 and 1 are primarily based on the percentage difference in OS at 2 years. However, our study underscores the importance of analysis based on RMST, demonstrating that at 2 years, the difference between the two trials is not dissimilar in terms of clinical benefit. In other words, the use of RMST not only allows for the meticulous evaluation of potential benefits, which may be limited in absolute terms, but also facilitates the comparison of clinical benefits between different trials at the same fixed time point. A key point in assessing clinical benefits is understanding whether treatment efficacy is balanced with safety and improvements in quality of life (QoL). In both RCTs, although QoL was not a primary endpoint, the observed improvement in efficacy did not correspond to an improvement in QoL [29,30]. Likewise, there were no significant differences in the occurrence of grade 3–4 adverse events, severe adverse events (grade 5), or events leading to treatment discontinuation between the treatment groups in both RCTs [8,19].

The RMST difference assessment may facilitate weighing the benefits against potential adverse effects, associated costs, or the impact of time toxicity [27]. Time toxicity, the concept of time spent coordinating care and attending frequent medical appointments (including travel and waiting times), emergency care for side effects, hospitalization, and time invested in follow-up tests, is usually undervalued when it comes to transferring effectiveness to clinical practice [31]. These considerations are not trivial, and in some cases, the time lost in receiving certain treatment regimens should be balanced with the survival gains offered by treatment [31]. Similarly, the contradictory results of different cost-effectiveness analyses further reinforce the potential benefits of assessing clinical benefits in absolute terms, as offered by RMST [32,33,34,35,36,37].

Our systematic review and meta-analysis have some limitations. First, we only had two RCTs, and because direct access to RCT data was not feasible, survival times and individual-level data were reconstructed by using published KM curves, which may have impaired data accuracy [27,38]. However, after reconstructing the curves and HRs, there was an agreement between the estimated and reported results (Appendix A).

Second, it is important to note that KM estimates often become imprecise towards the end of the OS curve, since the number of patients at risk is reduced [27]. Therefore, we used a specific time horizon of clinical relevance (24 months) and a sub-analysis was conducted based on the minimum longest observed event time in both the intervention and control groups (30 months).

Finally, KEYNOTE-966 and TOPAZ-1 share many similarities, but also have notable differences. One distinction lies in the duration of the gemcitabine treatment [8]. In TOPAZ-1, gemcitabine was limited to eight cycles, whereas in KEYNOTE-966, gemcitabine could be continued until disease progression or intolerable toxicity, with no set maximum number of cycles. The varying gemcitabine treatment durations between KEYNOTE-966 and TOPAZ-1 reflect differences in clinical practice, but undoubtedly do not represent a change in terms of OS benefit. Notably, KEYNOTE-966 employed stratified randomization by geographical region, whereas TOPAZ-1 did not [7,8]. Furthermore, KEYNOTE-966 enrolled a larger number of participants (1069 vs. 685 in TOPAZ-1), including a higher proportion from outside Asia (55% vs. 45%) [7,8]. Finally, KEYNOTE-966 and TOPAZ-1 exhibited variations in the shape of their OS curves and in the timing of their separation. In KEYNOTE-966, the curves began to favor the pembrolizumab group around the second month after randomization, maintaining a relatively steady separation throughout the study [8]. Conversely, in TOPAZ-1, the curves initially crossed, and the durvalumab group did not show a survival advantage until approximately six months post-randomization [19]. Despite these differences, the heterogeneity between both studies was low, suggesting that they are comparable when drawing conclusions. Nevertheless, considering that only two studies were included in the meta-analysis, the absence of heterogeneity in the results is not fully trustable.

## 5. Conclusions

In conclusion, our systematic review and meta-analysis confirms that immunotherapy-based regimens in combination with GemCis confer a statistically significant survival advantage in patients with locally advanced or advanced BTC and quantifies the clinical benefit in terms of OS and PFS, which should be balanced with individual patient factors, preferences, and potential risks. The use of RMST is not only methodologically sound in these scenarios, but also facilitates understanding and communication with patients. The applicability of RMST in future RCTs with time-to-event outcomes should be encouraged because it offers valuable insights into treatment effects and complements the usual measures of relative and absolute risk reduction. This additional information might be useful for healthcare agencies in the process of drug approval and/or reimbursement and for patients and clinicians in treatment decision-making in the setting of value-based medicine.

## Figures and Tables

**Figure 1 cancers-16-02077-f001:**
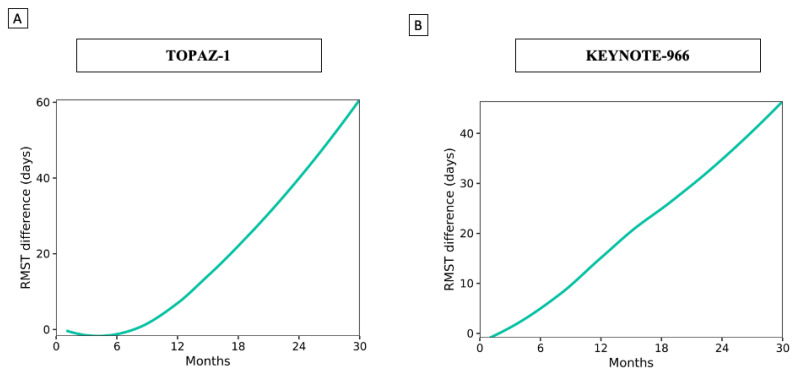
Variation in RMST difference over time. (**A**) TOPAZ-1 trial; (**B**) KEYNOTE-966 trial.

**Figure 2 cancers-16-02077-f002:**
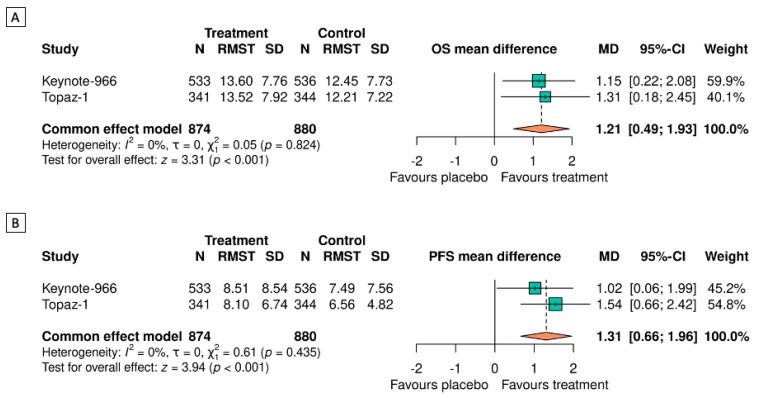
Meta-analysis of the mean differences in OS (**A**) and PFS (**B**) at 24 months by RMST.

**Figure 3 cancers-16-02077-f003:**
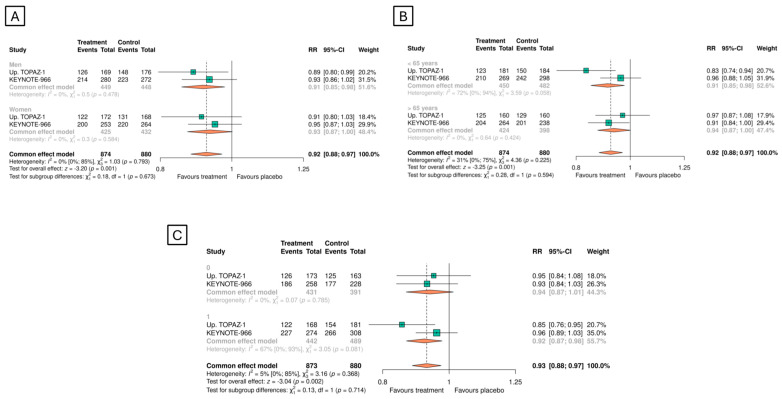
Overall survival in the predefined subgroups: (**A**) sex, (**B**) age, and (**C**) ECOG.

**Figure 4 cancers-16-02077-f004:**
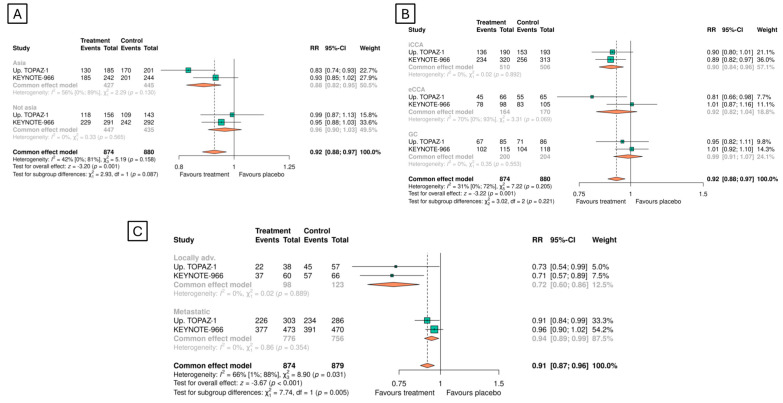
Overall survival in the predefined subgroups: (**A**) region, (**B**) primary tumor location, and (**C**) stage.

**Table 1 cancers-16-02077-t001:** Outcomes reported in systemic treatment-positive phase 3 trials.

	TOPAZ-1		KEYNOTE-966	
	Durva + GemCis (n = 341)	GemCis (n = 344)	Pembro + GemCis (n = 533)	GemCis (n = 536)
Inclusion Criteria	- Aged 18 years or older. - Histologically confirmed unresectable locally advanced or metastatic CCA. - Disease measurable per RECIST version 1.1 determined by the investigator. - ECOG performance status of 0 or 1. - The only previous systemic therapy permitted was neoadjuvant or adjuvant therapy completed at least 6 months before the diagnosis of unresectable or metastatic disease.		- Aged 18 years or older. - Histologically confirmed unresectable locally advanced or metastatic CCA. - Disease measurable per RECIST version 1.1 determined by the investigator. - ECOG performance status of 0 or 1. - The only previous systemic therapy permitted was neoadjuvant or adjuvant therapy completed at least 6 months before the diagnosis of unresectable or metastatic disease.	
Stratified	Diseases status (initially unresectable vs. recurrent). Primary tumour location (iCCA vs. eCCA vs. GC).		Geographic region (Asia vs. not Asia). Disease stage (locally advanced vs. metastatic). Primary tumor location (iCCA vs. eCCA vs. GC).	
Age	64 (20–84)	64 (31–85)	64 (57–71)	63 (55–70)
Female Sex	50.4%	48.8%	47%	49%
Population (percentage)				
- Intrahepatic (iCCA)	56%	56%	60%	58%
- Extrahepatic (eCCA)	19%	19%	18%	20%
- Gallbladder (GC)	25%	25%	22%	22%
- Asia	52.2%	57%	45%	46%
Follow-up in months, median (IQR)	23.4 (20.6–25.2)	22.4 (21.4–23.8)	25.6 (21.7–30.4)	25.6 (21.7–30.4)
OS (months), median (95%CI)	12.9 (11.6–14.1)	11.3 (10.1–12.5)	12.7 (11.5–13.6)	10.9 (9.9–11.6)
PFS (months), median (95%CI)	7.2 (6.7–7.4)	5.7 (5.6–6.7)	6.5 (5.7–6.9)	5.6 (5.1–6.6)
ORR (percentage)	26.7%	18.7%	29%	29%
DCR (percentage)	85.3%	82.6%	75%	76%
AE grade 3–4 (percentage)	74%	75.1	79%	75%
Im-AE grade 3–4 (percentage)	2.4%	-	7%	-

GemCis: gemcitabine and cisplatin; CCA: cholangiocarcinoma; RECIST: Response Evaluation Criteria in Solid Tumors; ECOG: Eastern Cooperative Oncology Group; OS: overall survival; PFS: progression-free survival; ORR: objective response rate; DCR: disease control rate; AE: adverse events; Im-AE: immune-mediated adverse events.

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
