# Peer review of "Evaluation of Overall Survival by Restricted Mean Survival Time of Advanced Biliary Tract Cancer treated with Immunotherapy: A Systematic Review and Meta-Analysis"

_cancers, 2024, doi:10.3390/cancers16112077_

Round 1

Reviewer 1 Report

Comments and Suggestions for Authors

This is an interesting paper. Also evaluate the RMST at 12 months and discuss the differences between the RMST at 12 and 24 months.

Author Response

Barcelona, May 25th, 2024

To: Dr. Ilyas Sahin - Guest Editor “Recent Advances in Hepatobiliary Cancers: From Diagnosis to Treatment (Volume II)” - Cancers.

Dear colleagues

Point-by-point Response to Cancers-3011830

We wish to thank the editorial team and reviewers for their thorough review and the opportunity to resubmit our manuscript with significant revisions. We have carefully addressed the concerns highlighted by the reviewers, which we believe have significantly enhanced the quality of our manuscript.

We are optimistic that our revised manuscript is a strong candidate for publication in Cancers.

Below, we provide detailed responses to each of the reviewers' comments.

Reviewer #1: This is an interesting paper. Also evaluate the RMST at 12 months and discuss the differences between the RMST at 12 and 24 months.

R: Thank you for your comment and insightful suggestions. In Figure 1, we have plotted the survival benefit evaluated using RMST over time, and we have added a paragraph to the results explaining this outcome.

In the TOPAZ-1 study, the RMST difference started near zero and gradually increased after 6 months, reaching approximately 60 days by the end of the 30-month period, indicating a progressively growing survival benefit of the treatment compared to the control group. Similarly, in the KEYNOTE-966 study, the RMST difference started near zero and steadily increased, reaching approximately 40 days by the end of 30 months, suggesting a survival benefit of the treatment, albeit to a lesser extent than in TOPAZ-1. Overall, the RMST difference was greater in TOPAZ-1 than in KEYNOTE-966 over 30 months, but these differences may be attributed to variations in the study populations, treatments evaluated, or study designs.

Reviewer 2 Report

Comments and Suggestions for Authors

This systematic review is well-conducted and informative. I don't detect any critical issues. The only suggestion that I would like to make is to increase the resolution of Figs 3 and 4 because the writing is difficult to read.

Author Response

Barcelona, May 25th, 2024

To: Dr. Ilyas Sahin - Guest Editor “Recent Advances in Hepatobiliary Cancers: From Diagnosis to Treatment (Volume II)” - Cancers.

Dear colleagues

Point-by-point Response to Cancers-3011830

 We wish to thank the editorial team and reviewers for their thorough review and the opportunity to resubmit our manuscript with significant revisions. We have carefully addressed the concerns highlighted by the reviewers, which we believe have significantly enhanced the quality of our manuscript.

We are optimistic that our revised manuscript is a strong candidate for publication in Cancers.

Below, we provide detailed responses to each of the reviewers' comments.

Reviewer #2: This systematic review is well-conducted and informative. I don't detect any critical issues. The only suggestion that I would like to make is to increase the resolution of Figs 3 and 4 because the writing is difficult to read.

R: Thank you for your review and suggestions. We have updated the images for better readability and quality.

Reviewer 3 Report

Comments and Suggestions for Authors

The manuscript revealed the overall survival in GemCis plus durvalumb/pembrolizumab and GemCis treatment by RMST analysis, which is useful to improve the OS in advanced biliary tract cancer. Here are my concerns that could enhance the study.

Authors showed the OS and PFS based on different factors included sex, age, geography, stage and ECOG. How about the population, such as intrahepatic and gall bladder? Are they similar?

The meta-analysis is based on the RMST for the treatment effect. However, the hazard ratio (HR) is commonly used as another method. Do authors compare this two approaches in this analysis to show the discrepancy or to being a supporting evidence for authors' clarification?

Comments on the Quality of English Language

English language issue is minor for this manuscript.

Author Response

Barcelona, May 25th, 2024

To: Dr. Ilyas Sahin - Guest Editor “Recent Advances in Hepatobiliary Cancers: From Diagnosis to Treatment (Volume II)” - Cancers.

Dear colleagues

Point-by-point Response to Cancers-3011830

We wish to thank the editorial team and reviewers for their thorough review and the opportunity to resubmit our manuscript with significant revisions. We have carefully addressed the concerns highlighted by the reviewers, which we believe have significantly enhanced the quality of our manuscript.

We are optimistic that our revised manuscript is a strong candidate for publication in Cancers.

Below, we provide detailed responses to each of the reviewers' comments.

Reviewer #3: The manuscript revealed the overall survival in GemCis plus durvalumb/pembrolizumab and GemCis treatment by RMST analysis, which is useful to improve the OS in advanced biliary tract cancer. Here are my concerns that could enhance the study.

Authors showed the OS and PFS based on different factors included sex, age, geography, stage and ECOG. How about the population, such as intrahepatic and gall bladder? Are they similar?

The meta-analysis is based on the RMST for the treatment effect. However, the hazard ratio (HR) is commonly used as another method. Do authors compare these two approaches in this analysis to show the discrepancy or to being a supporting evidence for authors' clarification?

R: Thank you for your comments.

In Figure 4, we meta-analyzed overall survival data in the predefined subgroups: (A) region, (B) primary tumor location (iCCA, eCCA, and GC), and (C) stage (locally advanced and metastatic). The results suggest a consistent benefit in terms of OS (RR < 1) across all subgroups analyzed, although there was some heterogeneity in the analysis by cancer type and tumor stage. We have included this in the Results section of the predefined subgroup analysis.

Finally, the objective of using RMST aligns with the need for a robust method to evaluate outcomes in the absence of proportional hazards, which is common in the context of immunotherapy. We consider that the use of RMST is not only methodologically sound in these scenarios, but also facilitates understanding and communication with patients, improving decision-making, and reinforcing the concept of value-based medicine.

We have modified part of the Discussion to emphasize that this method of time-to-event trial evaluation is a complementary and robust tool that should be considered in clinical trials involving immunotherapy.
